# Enzymatic Synthesis of Unnatural Ginsenosides Using a Promiscuous UDP-Glucosyltransferase from *Bacillus subtilis*

**DOI:** 10.3390/molecules23112797

**Published:** 2018-10-28

**Authors:** Ting-Ting Zhang, Ting Gong, Zong-Feng Hu, An-Di Gu, Jin-Ling Yang, Ping Zhu

**Affiliations:** 1State Key Laboratory of Bioactive Substance and Function of Natural Medicines, Institute of Materia Medica, Peking Union Medical College & Chinese Academy of Medical Sciences, Beijing 100050, China; tingtingzhang_pumc@163.com (T.-T.Z.); gongting@imm.ac.cn (T.G.); huzongfeng@imm.ac.cn (Z.-F.H.); gad@imm.ac.cn (A.-D.G.); 2Key Laboratory of Biosynthesis of Natural Products of National Health Commission, Institute of Materia Medica, Peking Union Medical College & Chinese Academy of Medical Sciences, Beijing 100050, China

**Keywords:** UDP-glycosyltransferase, *Bacillus subtilis*, glycosylation, unnatural ginsenoside, enzymatic synthesis

## Abstract

Glycosylation, which is catalyzed by UDP-glycosyltransferases (UGTs), is an important biological modification for the structural and functional diversity of ginsenosides. In this study, the promiscuous UGT109A1 from *Bacillus subtilis* was used to synthesize unnatural ginsenosides from natural ginsenosides. UGT109A1 was heterologously expressed in *Escherichia coli* and then purified by Ni-NTA affinity chromatography. Ginsenosides Re, Rf, Rh1, and R1 were selected as the substrates to produce the corresponding derivatives by the recombinant UGT109A1. The results showed that UGT109A1 could transfer a glucosyl moiety to C3-OH of ginsenosides Re and R1, and C3-OH and C12-OH of ginsenosides Rf and Rh1, respectively, to produce unnatural ginsenosides 3,20-di-*O*-β-d-glucopyranosyl-6-*O*-[α-l-rhamnopyrano-(1→2)-β-d-glucopyranosyl]-dammar-24-ene-3β,6α,12β,20*S*-tetraol (**1**), 3,20-di-*O*-β-d-glucopyranosyl-6-*O*-[β-d-xylopyranosyl-(1→2)-β-d-glucopyranosyl]-dammar-24-ene-3β,6α,12β,20*S*-tetraol (**6**), 3-*O*-β-d-glucopyranosyl-6-*O*-[β-d-glucopyranosyl-(1→2)-β-d-glucopyranosyl]-dammar-24-ene-3β,6*α*,12β,20*S*-tetraol (**3**), 3,12-di-*O*-β-d-glucopyranosyl-6-*O*-[β-d-glucopyranosyl-(1→2)-β-d-glucopyranosyl]-dammar-24-ene-3β,6*α*,12β,20*S*-tetraol (**2**), 3,6-di-*O*-β-d-glucopyranosyl-dammar-24-ene-3β,6α,12β,20S-tetraol (**5**), and 3,6,12-tri-*O*-β-d-glucopyranosyl-dammar-24-ene-3β,6α,12β,20S-tetraol (**4**). Among the above products, **1**, **2**, **3**, and **6** are new compounds. The maximal activity of UGT109A1 was achieved at the temperature of 40 °C, in the pH range of 8.0–10.0. The activity of UGT109A1 was considerably enhanced by Mg^2+^, Mn^2+^, and Ca^2+^, but was obviously reduced by Cu^2+^, Co^2+^, and Zn^2+^. The study demonstrated that UGT109A1 was effective in producing a series of unnatural ginsenosides through enzymatic reactions, which could pave a way to generate promising leads for new drug discovery.

## 1. Introduction

Ginseng (*Panax ginseng* Meyer) has been widely used in traditional herbal medicine for thousands of years in Asia [1]. Ginsenosides, the principal active constituents of ginseng, have demonstrated a wide spectrum of pharmacological activities, including antitumor, anti-oxidation, anti-aging, and anti-inflammatory effects, and so on [2,3,4,5,6]. Most ginsenosides are glycosylated from the tetracyclic triterpenoid aglycones, protopanaxadiol (PPD), and protopanaxatriol (PPT) [7]. Glycosylation, which is catalyzed by UDP-glycosyltransferase (UGT), plays an important role in ginsenoside biosynthesis. Variation in the number, position, and type of sugar moieties attached to ginsenosides can markedly increase their structural diversity, and thus lead to changes in their bioactivities [8,9,10].

There are mainly two groups of ginsenosides, PPD-type and PPT-type, which are differentiated by the hydroxylation pattern. UGTs glycosylate PPD at C3-OH and/or C20-OH, and PPT at C6-OH and/or C20-OH. So far, the UGTs which catalyze the glycosylation of C3-OH, C6-OH, and C20-OH of PPD and PPT have all been identified from *P. ginseng*, respectively [11,12,13]. Although both PPD and PPT have a hydroxyl group at the C12 position, the C12-glycosylated ginsenosides have rarely been isolated and the UGT responsible for the glycosylation of C12-OH has never been identified from *Panax* species.

Microbes have many UGTs which can use some small molecules as substrates to produce their derivatives [14,15,16]. Among these microbial UGTs, some UGTs from *Bacillus* species have been widely used in the glycosylation of natural products, such as ginsenosides and flavonoids [17,18,19]. In our previous study, a new UGT named UGT109A1 from *B. subtilis* was demonstrated to catalyze the glycosylation of dammarenediol-II (DM) at C3-OH and C20-OH, and PPD and PPT at C3-OH and C12-OH, respectively, to produce unnatural ginsenosides [20]. Recently, another UGT Bs-YjiC was also identified from *B. subtilis* to transfer a glucosyl moiety to C3-OH and C12-OH of PPD, and C3-OH, C6-OH, and C12-OH of PPT, respectively [21,22].

Ginsenosides Re, Rf, Rh1, and R1 are all bioactive PPT-type ginsenosides, which also have free C3-OH and C12-OH [23,24,25,26,27]. Herein, UGT109A1 was heterologously expressed in *Escherichia coli* and purified by Ni-NTA affinity chromatography. Then, ginsenosides Re, Rf, Rh1, and R1 were selected as catalytic substrates to produce unnatural ginsenosides for drug leads. Furthermore, the reaction conditions of UGT109A1 were optimized and the effect of metal ions on this enzyme was also studied.

## 2. Results

### 2.1. Heterologous Expression and Purification of UGT109A1

The UGT109A1 gene was cloned from *B. subtilis* CTCC 63501 and then heterologously expressed in *E. coli* BL21 (DE3) with an N-terminal His6-tag. The recombinant UGT109A1 was expressed as a soluble protein and thus purified easily by Ni-NTA affinity chromatography. A total of 20 mM imidazole was used to get rid of protein impurity. Most of the recombinant UGT109A1 was eluted by 50 mM imidazole. The calculated molecular weight of the recombinant UGT109A1 was about 61 kDa, which corresponded with the result detected by SDS-PAGE (Figure 1).

### 2.2. In Vitro Glycosylation of Ginsenosides by UGT109A1

In this study, ginsenosides Re, Rf, Rh1, and R1 were used as substrates to synthesize novel glycosylation derivatives by the recombinant UGT109A1. After incubation for 12 h at 37 °C, the reactants were detected by HPLC analysis. HPLC analysis indicated that the recombinant UGT109A1 catalyzed ginsenoside Re to product **1** with a conversion rate of 71.4% (Figure 2A), ginsenoside Rf to products **2** with a conversion rate of 73.2%, and **3** with a conversion rate of 18.7% (Figure 2B), ginsenoside Rh1 to products **4** with a conversion rate of 20.9% and **5** with a conversion rate of 32.1% (Figure 2C), and ginsenoside R1 to product **6** with a conversion rate of 49.7% (Figure 2D).

### 2.3. Structure Identification of the Glycosylated Products

The structures of the glycosylated products of ginsenosides Re, Rf, Rh1, and R1 catalyzed by UGT109A1 were determined by HR-ESI-MS, 1D-NMR (^1^H-NMR and ^13^C-NMR), and 2D-NMR (HMBC and HSQC) analysis.

The HR-ESI-MS analysis (*m*/*z* 1131.5965 [M + Na]^+^) of product **1** showed that the molecular formula of product **1** was C_54_H_92_O_23_ (Figure 3A). The ^1^H-NMR and ^13^C-NMR data suggested that product **1** was the glycosylated derivative, with one additional glucosyl moiety attached to the substrate ginsenoside Re (Appendix A and Appendix A). The observation of a significant ^13^C downfield shift of 12.2 ppm of C3 indicated that the additional glucosyl moiety was linked to the C3 position of ginsenoside Re, which was supported by the HMBC correlations of sugar anomeric signal H-1′′′′ (δ_H_ 4.97) with C-3 (δ_C_ 89.6) and H-3 (δ_H_ 3.38, dd, *J* = 13.2, 4.8 Hz) with C-1′′′′ (δ_C_ 107.6). Thus, product **1** was determined to be 3,20-di-*O*-β-d-glucopyranosyl-6-*O*-[*α*-l-rhamnopyranosyl-(1→2)-β-d-glucopyranosyl]-dammar-24-ene-3β,6α,12β,20S-tetraol, which was a new compound (Figure 4A).

The molecular formulas of products **2** and **3** were established as C_54_H_92_O_24_ and C_48_H_82_O_19_, respectively, based on HR-ESI-MS data (*m*/*z* 1147.5919 [M + Na]^+^ and *m*/*z* 985.5394 [M + Na]^+^) (Figure 3B,C). The HR-ESI-MS data suggested that product **2** was the glycosylated derivative, with two additional glucosyl moieties attached to the substrate ginsenoside Rf. The ^1^H-NMR and ^13^C-NMR data of product **2** were similar to those of ginsenoside Rf, except for two additional hexose sugars (Appendix A and Appendix A).

In the HMBC spectrum of product **2**, the correlations of H-1′′′ (δ_H_ 5.03, d, *J* = 7.5 Hz) with C-3 (δ_C_ 90.2), H-3 (δ_H_ 3.39, dd, *J* = 11.5, 4.0 Hz) with C-1′′′ (δ_C_ 107.8), H-1′′′′ (δ_H_ 5.28, d, *J* = 7.5 Hz) with C-12 (δ_C_ 79.1), and H-12 (δ_H_ 3.81) with C-1′′′′ (δ_C_ 100.6) suggested that two glucosyl moieties were located at C3 and C12 positions of ginsenoside Rf. The anomeric β-configurations were supported by the large coupling constant (J = 7.5 Hz). Hence, product 2 was elucidated to be 3,12-di-O-β-d-glucopyranosyl-6-O-[β-d-glucopyranosyl-(1→2)-β-d-glucopyranosyl]-dammar-24-ene-3β,6α,12β,20S-tetraol (Figure 4B).

The ^1^H-NMR and ^13^C-NMR data of product **3** showed that there was an additional hexose sugar compared with that of ginsenoside Rf (Appendix A and Appendix A). The HMBC correlations of H-1′′′ (δ_H_ 5.02, d, *J* = 7.8 Hz) with C-3 (δ_C_ 90.3) and H-3 (δ_H_ 3.43, dd, *J* = 12.0, 4.2 Hz) with C-1′′′ (δ_C_ 107.7) showed that the glucosyl moiety was located at the C3 position of ginsenoside Rf. The anomeric β-configuration was deduced through analysis of the large coupling constant of H-1′′′ (*J* = 7.8 Hz). These results indicated that product **3** was 3-*O*-β-d-glucopyranosyl-6-*O*-[β-d-glucopyranosyl-(1→2)-β-d-glucopyranosyl]-dammar-24-ene-3β,6α,12β,20*S*-tetraol (Figure 4B). Products **2** and **3** were both new compounds.

Products **4** and **5** had the molecular formulas C_48_H_82_O_19_ and C_42_H_72_O_14_, respectively, deduced by the molecular ion peaks in the HR-ESI-MS analysis (*m*/*z* 985.5330 [M + Na]^+^ and *m*/*z* 823.4825 [M + Na]^+^) (Figure 3D,E). The ^1^H-NMR and ^13^C-NMR spectra of product **4** suggested that there were two additional glucosyl moieties attached to ginsenoside Rh1 (Appendix A and Appendix A). Furthermore, the HMBC correlations of the sugar anomeric signals H-1′′ (δ_H_ 4.29, d, *J* = 8.0 Hz) with C3 (δ_C_ 90.9), H-3 (δ_H_ 3.22) with C-1′′ (δ_C_ 107.0), H-1′′′ (δ_H_ 4.47, d, *J* = 8.0 Hz) with C12 (δ_C_ 79.4), and H-12 (δ_H_ 3.23) with C-1′′′ (δ_C_ 100.6) demonstrated that product **4** was 3,6,12-Tri-*O*-β-d-glucopyranosyl-dammar-24-ene-3β,6α,12β,20*S*-tetraol (Figure 4C). Product **5** was determined to be 3,6-Di-*O*-β-d-glucopyranosyl-dammar-24-ene-3β,6α,12β,20S-tetraol (Figure 4C) based on its 1D- and 2D-NMR data (Appendix A and Appendix A). The HMBC correlations of the sugar anomeric signal H-1′′ (δ_H_ 4.32, d, *J* = 7.5 Hz) with C3 (δ_C_ 90.9) and H-3 (δ_H_ 3.27) with C-1′′ (δ_C_ 107.0) suggested that a β-glucosyl moiety was attached to the C3 position of ginsenoside Rh1.

The molecular formula of product **6** was determined to be C_53_H_90_O_23_ based on HR-ESI-MS data (*m*/*z* 1117.5222 [M + Na]^+^) (Figure 3F). The ^1^H-NMR and ^13^C-NMR data were similar to those of ginsenoside R1, expect for additional glucosyl moiety signals of **6** (Appendix A and Appendix A). The ^13^C-NMR analysis showed a significant downfield shift of 11.7 ppm of C-3 (δ_C_ 90.1) compared with that of ginsenoside R1. The HMBC correlations from the sugar anomeric signal H-1′′′′ (δ_H_ 5.02, d, *J* = 7.8 Hz) to C-3 (δ_C_ 90.1), and from H-3 (δ_H_ 3.45, dd, *J* = 11.4, 4.2 Hz) to C-1′′′′ (δ_C_ 107.6), confirmed that the glucosyl moiety was linked to the C3 position of ginsenoside R1. The structure of product **6** was elucidated to be 3,20-Di-*O*-β-d-glucopyranosyl-6-*O*-[β-d-xylopyranosyl-(1→2)-β-d-glucopyranosyl]-dammar-24-ene-3β,6α,12β,20*S*-tetraol, which was a new compound (Figure 4D).

### 2.4. Optimization of Reaction Conditions for UGT109A1

To investigate the optimal reaction conditions, we examined the effects of temperature, pH, and metal ions on UGT109A1 activity. The relationship between enzymatic activity and temperature was determined in the range of 20–60 °C. Similar to other UGTs from *B. subtilis* [17,19], UGT109A1 showed its maximal activity at 40 °C (Figure 5). UGT109A1 showed activity in the pH range of 4.0–11.0, with its maximal activity from pH 8.0 to pH 10.0 (Figure 6). The effect of metal ions on UGT109A1 activity was investigated at 40 °C and pH 8.0–10.0. UGT109A1 activity appeared to be considerably enhanced by Mg^2+^, Mn^2+^, and Ca^2+^, but obviously reduced by Cu^2+^, Co^2+^, and Zn^2+^. Furthermore, Mn^2+^ exhibited the most significant promotion effect, which could even increase UGT109A1 activity by eight times when ginsenoside Rh1 was used as a substrate (Figure 7).

## 3. Discussion

Glycosylation catalyzed by UGTs is usually the last step of natural product biosynthesis, but it plays a vital role in generating bioactive natural compounds [28]. In the process of ginsenoside biosynthesis, UGTs catalyze the transfer of glycosyl residues from activated sugars to the aglycones, and thus regulate properties of ginsenosides, such as bioactivity, solubility, and stability. Moreover, glycosylation also determines the diversity of ginsenosides. Up to now, several UGTs have been characterized in *P. ginseng*, which exhibit strict substrate selectivity and specificity [11,12,13]. It has been reported that UGTs from microbes possess a more open and larger acceptor binding pocket than those from plants and therefore endow themselves with broad substrate specificity [29,30,31]. Therefore, research on microbial UGTs has attracted extensive interest and made great progress in the enzymatic glycosylation of natural and unnatural products. Accordingly, we identified a new UGT named UGT109A1 from *B. subtilis* and then heterologously expressed in *E. coli* BL21 (DE3). In our previous study, we have demonstrated that recombinant UGT109A1 can transfer a glucose moiety to C3-OH and C20-OH of DM, and C3-OH and C12-OH of PPD and PPT, respectively, to produce unnatural ginsenosides, showing excellent enzyme promiscuity [20]. Another UGT Bs-YjiC from *B. subtilis* was proved to transfer a glucosyl moiety to C3-OH and C12-OH of PPD, and C3-OH, C6-OH, and C12-OH of PPT [21,22]. The amino acid sequence of UGT109A1 has 94.9% identity with Bs-YjiC. Bs-YjiC can transfer a glucose moiety to C6-OH of PPT, while UGT109A1 cannot. This evidence indicates that UGT109A1 is different from Bs-YjiC not only in the amino acid sequence, but also in the enzyme function.

Increasing the sugar moiety number of ginsenosides can enrich the structural diversity of ginsenosides and perhaps leads to changes in their bioactivities. However, research on using UGT109A1 to further glycosylate ginsenosides has never been reported before. Ginsenosides Re, Rf, Rh1, and R1 are all PPT-type ginsenosides with free C3-OH and C12-OH, which exhibit diverse pharmacological activities. Ginsenoside Re functions as an antioxidant, protecting cardiomyocytes from oxidant injury induced by both exogenous and endogenous oxidants [23], and protects MA-induced dopaminergic neurotoxicity [24]. Ginsenoside Rf was a promising novel therapeutic agent for the prevention and treatment of gastrointestinal inflammatory diseases [25]. Ginsenoside Rh1 exhibited potent characteristics of anti-inflammatory, antioxidant, and immunomodulatory effects, and positive effects on the nervous system [26]. Ginsenoside R1 can reduce the area of cerebral infarction, reduce apoptosis of hippocampal neurons, and protect rats from cerebral ischemia-reperfusion injury [27]. In the current study, ginsenosides Re, Rf, Rh1, and R1 were used as substrates to biosynthesize unnatural ginsenosides by the catalysis of UGT109A1. The results showed that UGT109A1 could transfer a glucose moiety to C3-OH of ginsenosides Re and R1, and C3-OH and C12-OH of ginsenosides Rf and Rh1, respectively, producing six unnatural ginsenosides. Among them, products **1**, **2**, **3**, and **6** were new compounds; products **4** and **5** were previously biosynthesized by glycosyltransferase Bs-Yjic [22]. Both ginsenosides Re and R1 have free C12-OH, but no C12-glycosylated products were detected in their reactions catalyzed by UGT109A1, which might be attributed to the steric hindrance generated by C20-glucosyl. Both ginsenosides Rf and Rh1 have free C20-OH, but no C20-glycosylated products were detected in their reactions, perhaps due to the fact that the catalytic efficiency of UGT109A1 for the C12-OH glycosylation was much higher than that for the C20-OH glycosylation, as reported by our previous study [20].

In order to synthesize unnatural ginsenosides efficiently, we studied the optimal reaction conditions of UGT109A1 and confirmed that the maximal activity of UGT109A1 was achieved at the temperature of 40 °C, in the pH range of 8.0–10.0. Moreover, we found that the activity of UGT109A1 could be considerably enhanced by Mg^2+^, Mn^2+^, and Ca^2+^, but obviously reduced by Cu^2+^, Co^2+^, and Zn^2+^. However, Mg^2+^, Mn^2+^, and Ca^2+^ dramatically activated the enzymatic activity of UGT109A1 toward ginsenoside Rh1, while they only activated the activity of UGT109A1 toward ginsenoside Re a little. A possible explanation for this might be that the effect of these metal ions on the affinity of UGT109A1 toward ginsenoside Rh1 was higher than that of UGT109A1 toward ginsenoside Re. The results of the metal ion effect are essentially in agreement with the previous findings on the two other UGTs from *B. subtilis*, except that Co^2+^ greatly improved the activities of the two other UGTs [17,32].

In summary, we have investigated the sugar acceptor promiscuity and the optimal reaction conditions of UGT109A1, which could pave a way for the mass production of unnatural ginsenosides. The results suggest that a diversity of unusual ginsenosides can be generated through the enzymatic synthesis of UGT109A1, and thus demonstrate its application prospect in finding promising leads for new drug discovery. The increase of glucose moieties may enhance the solubility, stability, and bioactivity of ginsenosides Re, Rf, Rh1, and R1. The pharmacological activities of the unnatural ginsenosides synthesized in this study will be investigated next. In order to generate more unnatural ginsenosides, the sugar donor promiscuity of UGT109A1 for glycosylation should be further studied. Furthermore, it is necessary to conduct a structural study of UGT109A1 to elucidate its structure–function relationship in the future.

## 4. Materials and Methods

### 4.1. Materials

Ginsenosides Re, Rf, Rh1, and R1 were purchased from Nanjing Spring & Autumn Biological Engineering Company (Nanjing, China). UDP-glucose (UDPG) was purchased from Sigma-Aldrich (St. Louis, MO, USA). All other chemicals and reagents are of an analytical grade.

### 4.2. Gene Cloning, Expression and Purification of UGT109A1

The UGT109A1 gene was cloned from *B. subtilis* CTCC 63501 and then heterologously expressed in *E. coli* BL21 (DE3) with an N-terminal His_6_-tag, which was described in detail in our previous report [20]. The crude recombinant UGT109A1 was subjected to Ni-NTA affinity chromatography; eluted successively with 20 mM, 50 mM, and 100 mM imidazole; and finally confirmed by SDS-PAGE.

### 4.3. In Vitro Activity Assay of UGT109A1

An activity assay of UGT109A1 was performed in a 100 μL reaction mixture containing 1 mM substrate (gensinosides Re, Rf, Rh1, or R1) dissolved in dimethyl sulfoxide (DMSO), 4 mM UDPG, 50 μg purified UGT109A1, and 25 mM Tris-HCl buffer (pH 8.0). The reaction mixtures were incubated at 37 °C for 12 h and then terminated by adding 100 μL methanol. Subsequently, the reactants were centrifuged at 10,000× *g* for 20 min and filtered by a 0.22 μm filter prior to HPLC analysis.

### 4.4. Effects of Temperature, pH and Metal Ions on the Activity of UGT109A1

Ginsenosides Re, Rf, Rh1, and R1 were used as sugar acceptors to determine the optimal reaction conditions of UGT109A1. To study the optimal reaction temperature for UGT109A1, the enzyme activity was determined at various temperatures ranging from 20 °C to 60 °C in 25 mM Tris-HCl buffer (pH 8.0). The effect of pH on enzyme activity was determined in the pH range of 4.0–6.0 (HAc-NaAc), 7.0–9.0 (Tris-HCl), and 10.0–11.0 (Na_2_HPO_4_-NaOH). The reactions of different pH levels were performed at 40 °C. The effect of metal ions on enzyme activity was tested in the presence of 10 mM CuCl_2_, CoCl_2_, ZnCl_2_, MgCl_2_, MnCl_2_, and CaCl_2_, respectively, at optimal pH 8.0–10.0 and 40 °C. A reaction without the addition of metal ions was used as the control. All reactions were carried out with UDPG as a sugar donor and all data are presented as means ± SD of three independent repeats.

### 4.5. HPLC Analysis of the Glycosylated Products

HPLC analysis was performed on an Agilent 1200 HPLC system (Agilent Technologies Inc., Palo Alto, CA, USA). To analyze the glycosylated products of ginsenosides Re, Rf, Rh1, and R1, the reverse-phase C_18_ column (4.6 mm × 150 mm, 5 μm particles, CAPCELL PAK, Tokyo, Japan) was employed at 28 °C. The gradient elution was performed with distilled H_2_O (solvent A) and acetonitrile (solvent B) in the following gradients: 0–25 min, 25–85% B; 25–45 min, 85% B. The UV wavelength of 203 nm was used to analyze the substrates and their glycosylated products.

### 4.6. Scale-Up Reactions and Accumulation of the Glycosylated Products

Scale-up reactions have a total volume of 60 mL, containing 100 μg/mL purified UGT109A1, 25 mM Tris-HCl (pH 8.0), 10 mM substrate dissolved in DMSO, and 50 mM UDPG. The reactions were performed at 37 °C for 12 h and terminated by adding 60 mL methanol. Then, the reactants were evaporated under reduced pressure distillation. The remaining residues were resuspended in 5.0-10.0 mL methanol, filtered by a 0.22 μm filter, and then purified by semi-preparative HPLC. Semi-preparative HPLC was performed on an HPLC system equipped with a Shimadzu LC-6AD pump and a Shimadzu SPD-20A prominence UV-VIS detector (Shimadzu Corporation, Kyoto, Japan) using a C_18_ column (10 mm × 250 mm, 5 μm particles, CAPCELL PAK, Japan). The HPLC conditions are described in detail in Appendix A.

### 4.7. HPLC-ESI-MS and NMR Analysis of the Glycosylated Products

Chromatographic analysis and HR-ESI-MS data were acquired using the HPLC-LTQ/FTICR-MS system (Thermo Fisher Scientific, Waltham, MA, USA) equipped with a C_18_ column (4.6 mm × 150 mm, 5 μm particles, CAPCELL PAK, Japan). NMR experiments were performed in methnol-*d*_4_ or pyridine-*d*_5_ for all ginsenosides on an INOVA 500 or 600 MHz NMR spectrometer (Varian, Palo Alto, CA, USA) with reference to the solvent peaks.

## Figures and Tables

**Figure 1 molecules-23-02797-f001:**
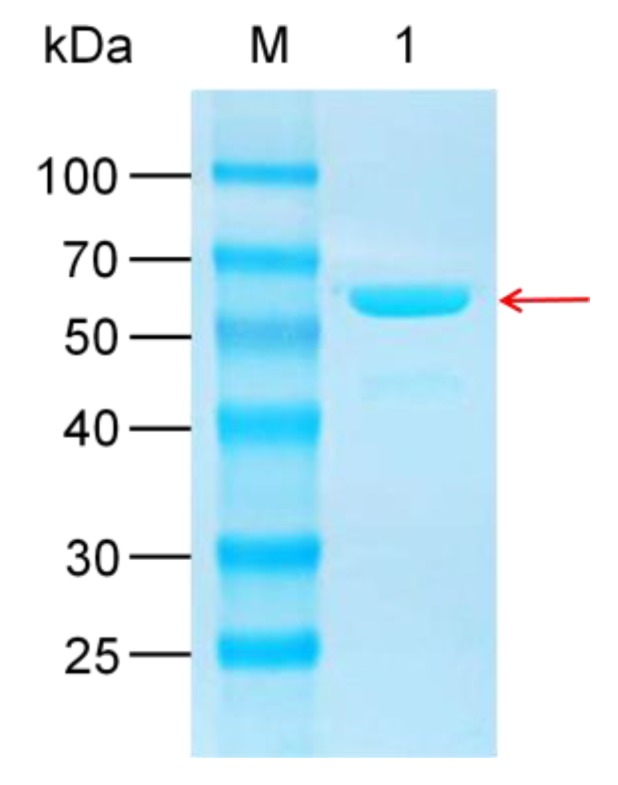
SDS-PAGE analysis of the recombinant UGT109A1 expressed in *Escherichia coli*. M: Molecular weight marker; 1: The recombinant UGT109A1 purified by Ni-NTA affinity chromatography.

**Figure 2 molecules-23-02797-f002:**
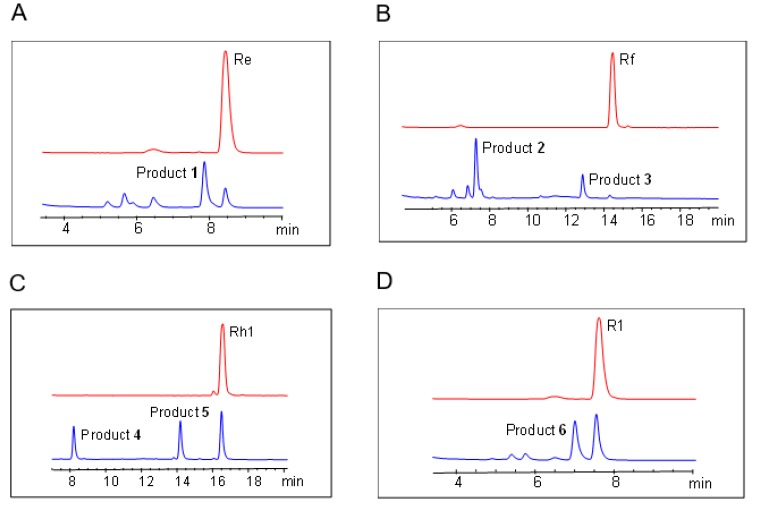
HPLC analysis of the glycosylated products catalyzed by UGT109A1. (**A**) Ginsenoside Re as a substrate; (**B**) Ginsenoside Rf as a substrate; (**C**) Ginsenoside Rh1 as a substrate; (**D**) Ginsenoside R1 as a substrate.

**Figure 3 molecules-23-02797-f003:**
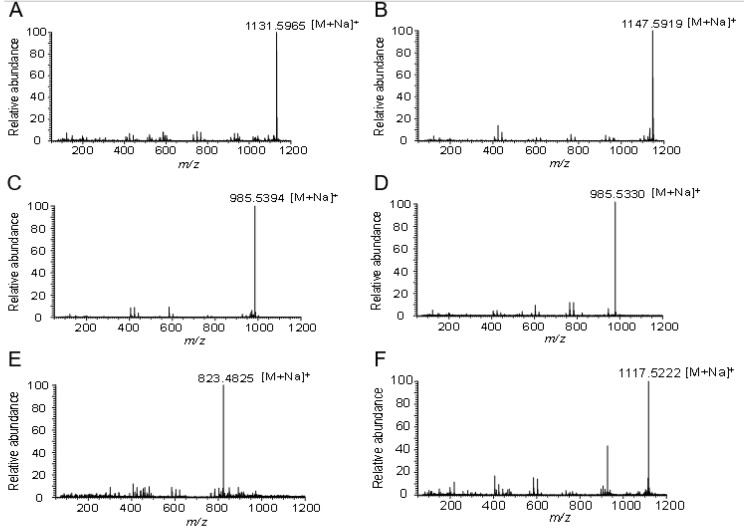
HR-ESI-MS analysis of the glycosylated products catalyzed by UGT109A1. (**A**) Product **1** synthesized from ginsenoside Re; (**B**) Product **2** synthesized from ginsenoside Rf; (**C**) Product **3** synthesized from ginsenoside Rf; (**D**) Product **4** synthesized from ginsenoside Rh1; (**E**) Product **5** synthesized from ginsenoside Rh1; (**F**) Product **6** synthesized from ginsenoside R1.

**Figure 4 molecules-23-02797-f004:**
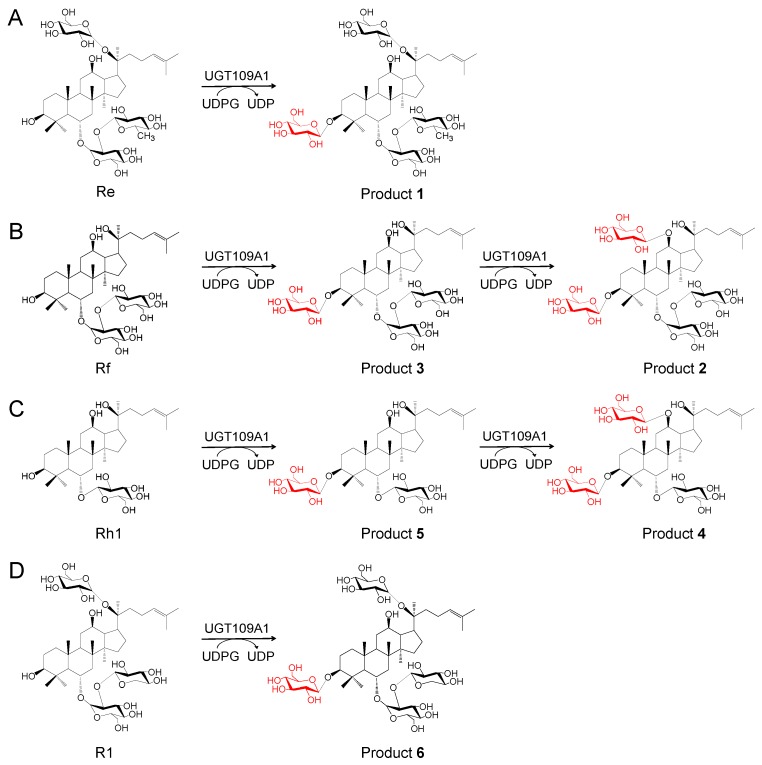
Glycosylation patterns of UGT109A1 toward ginsenosides Re (**A**), Rf (**B**), Rh1 (**C**), and R1 (**D**).

**Figure 5 molecules-23-02797-f005:**
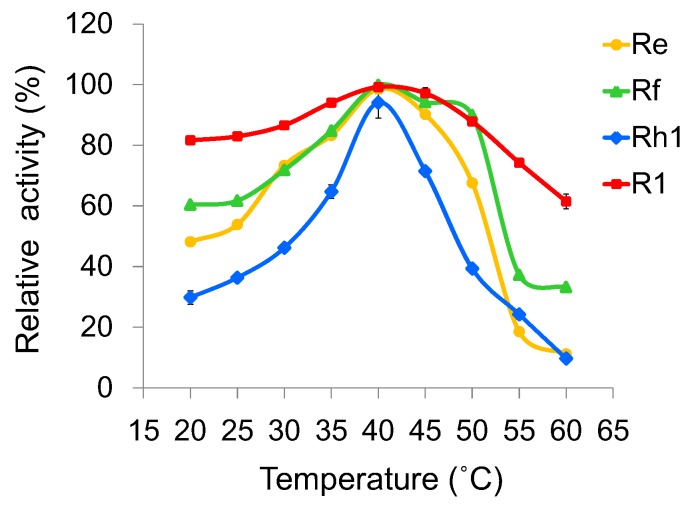
Effect of temperature on the enzymatic activity of UGT109A1.

**Figure 6 molecules-23-02797-f006:**
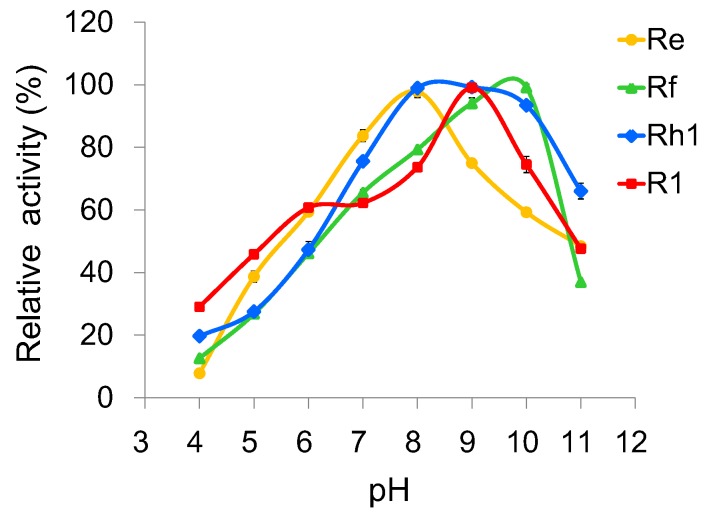
Effect of pH on the enzymatic activity of UGT109A1.

**Figure 7 molecules-23-02797-f007:**
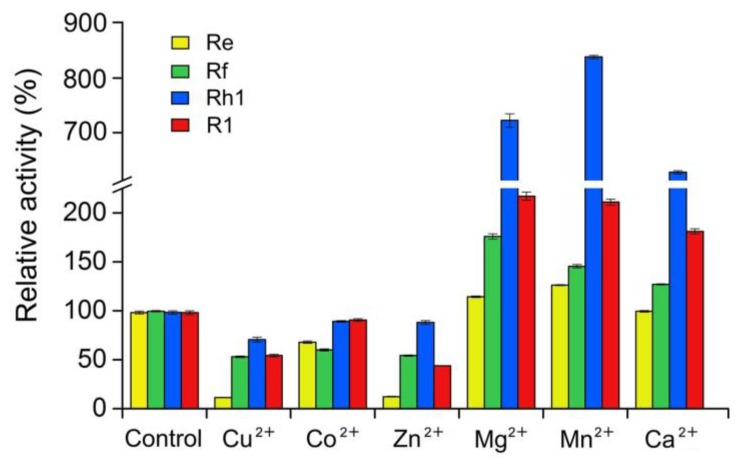
Effect of metal ions on the enzymatic activity of UGT109A1.

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
