# Peer review of "Enzymatic Synthesis of Unnatural Ginsenosides Using a Promiscuous UDP-Glucosyltransferase from Bacillus subtilis"

_molecules, 2018, doi:10.3390/molecules23112797_

Round 1

Reviewer 1 Report

The authors has added more discussions. I would like to recommend this manuscript to be accepted for publication in Molecules.

Author Response

Thank you very much for your recommendation for our manuscript to be accepted for publication in Molecules.

Reviewer 2 Report

This is a straightforward manuscript describing the modification of extant ginsenosides by a promiscuous UDP-glycosyltransferase (UGT) which can glycosylate positions C3, C20 and C12 of the parent ginsenoside. 

Glycosylation on C12 seems to be impeded by the presence of glycosyl residues on C20, and several metal ions are shown to affect catalysis to widely different extents, depending on the parent ginsenoside. Although the research described herein seems to have been carried out to a high standard I am left with the impression that the manuscript does not add too much to previous reports of the promiscuity of the UGT used. I would  have liked to see some attempt to explain why the dramatic increase of the catalytic rate upon use of Mg2+/Mn2+ (which I presume to be due to complexation of the diphoshate moiety in UDG-glucose) is not observed equally for all ginsenosides. I also think that an attempt to build a model of this UGT with the ginsenosides could have brought some more robustness to the suggestion that previous C20 glycosylation sterically impedes reaction on C12.

Author Response

Dear reviewer,

Thank you very much for your comment and suggestion.

1. In our manuscript, we found that only C12-OH can be glycosylated when ginsenosides have both free C12-OH and C20-OH simultaneously such as Rf and Rh1, perhaps due to the fact that the catalytic efficiency of UGT109A1 for the C12-OH glycosylation was much higher than that for the C20-OH glycosylation as reported by our previous study. But when ginsenosides have a glucosyl moiety at C20 position such as Re and R1, C12-OH cannot be glycosylated any longer. According to these results, we presumed that previous C20 glycosylation sterically impedes reaction on C12. In the current report, we have not explored the mechanism of UGT109A1. It is necessary to conduct structural study of UGT109A1 to elucidate its structure-function relationship in the future.

2. According to your suggestion, we have added some discussion to previous reports of the promiscuity of the UGT used, which has been highlighted in red color in the revised version.

3. Your presumption that the dramatic increase of the catalytic rate by use of Mg2+/Mn2+ is due to complexation of the diphoshate moiety in UDG-glucose may be reasonable. I have also tried to explain why the dramatic increase of the catalytic rate upon use of Mg2+/Mn2+ is not observed equally for all ginsenosides, which has been highlighted in red color in the revised version. 

Round 2

Reviewer 2 Report

I would have preferred to see solid data regarding the reasons for metal ion specificity and specificity of glycosilation position. but the manuscript can be accepted for publication. I think it does not fit "Molecules" stated "cutting-edge research" scope and it would be a much better fit for MDPI's  megajournal "J"